# The Main Protease of Middle East Respiratory Syndrome Coronavirus Induces Cleavage of Mitochondrial Antiviral Signaling Protein to Antagonize the Innate Immune Response

**DOI:** 10.3390/v16020256

**Published:** 2024-02-05

**Authors:** Mariska van Huizen, Xavier M. Vendrell, Heidi L. M. de Gruyter, A. Linda Boomaars-van der Zanden, Yvonne van der Meer, Eric J. Snijder, Marjolein Kikkert, Sebenzile K. Myeni

**Affiliations:** Molecular Virology Laboratory, Leiden University Center of Infectious Diseases (LU-CID), Leiden University Medical Center, 2333 ZA Leiden, The Netherlands

**Keywords:** MERS-CoV, M^pro^, immune evasion, MAVS

## Abstract

Mitochondrial antiviral signaling protein (MAVS) is a crucial signaling adaptor in the sensing of positive-sense RNA viruses and the subsequent induction of the innate immune response. Coronaviruses have evolved multiple mechanisms to evade this response, amongst others, through their main protease (M^pro^), which is responsible for the proteolytic cleavage of the largest part of the viral replicase polyproteins pp1a and pp1ab. Additionally, it can cleave cellular substrates, such as innate immune signaling factors, to dampen the immune response. Here, we show that MAVS is cleaved in cells infected with Middle East respiratory syndrome coronavirus (MERS-CoV), but not in cells infected with severe acute respiratory syndrome coronavirus 2 (SARS-CoV-2). This cleavage was independent of cellular negative feedback mechanisms that regulate MAVS activation. Furthermore, MERS-CoV M^pro^ expression induced MAVS cleavage upon overexpression and suppressed the activation of the interferon-β (IFN-β) and nuclear factor-κB (NF-κB) response. We conclude that we have uncovered a novel mechanism by which MERS-CoV downregulates the innate immune response, which is not observed among other highly pathogenic coronaviruses.

## 1. Introduction

In the last two decades, three coronaviruses (CoVs) that are highly pathogenic for humans have emerged, namely severe acute respiratory syndrome (SARS)-CoV in 2002, Middle East respiratory syndrome (MERS)-CoV in 2012, and SARS-CoV-2 in 2019. The diseases caused by these viruses are mainly characterized by respiratory symptoms, although other organs can also be affected [1,2]. Furthermore, severe disease is attributed to an exacerbated immune response that is characterized by increased infiltration of inflammatory monocytes, macrophages, and neutrophils in the lungs and a cytokine storm [3,4,5]. This dysregulated immune response could in part result from manipulation of the immune response by specific viral proteins [5,6,7]. The coronavirus genome encodes two large replicase polyproteins, pp1a and pp1ab, together consisting of 16 non-structural proteins (NSPs) that are primarily involved in organizing viral genome expression and replication [8,9,10]. These NSPs are released from the polyproteins by two internally encoded proteases, the papain-like protease (PL^pro^) in NSP3 and the main protease (M^pro^) in NSP5. Both proteases have previously been shown to manipulate various cellular processes, including the innate immune response [11,12,13,14,15]. In addition, the genome encodes the structural proteins and several accessory proteins, of which the latter also play an important role in pathogenesis and immune evasion [16,17].

The retinoic acid-inducible gene I (RIG-I)-like receptor (RLR) pathway is crucial in the defense against positive-sense RNA virus infection. The RLR family contains three members, namely RIG-I, melanoma differentiation-associated protein 5 (MDA5), and laboratory of genetics and physiology 2 (LGP2), which sense various species of double-stranded (ds)RNA [18]. All RLRs contain a C-terminal domain that prevents auto-activation and a helicase domain that recognizes specific immunostimulatory RNAs. In addition, RIG-I and MDA5 contain two caspase activation and recruitment (CARD) domains, which are exposed upon RNA binding and induce oligomerization of RIG-I/MDA5. These oligomers then translocate to the mitochondria, where they interact with mitochondrial antiviral signaling protein (MAVS) via CARD–CARD interactions to induce its oligomerization. In addition to a CARD domain, MAVS contains a transmembrane domain that anchors it to the mitochondrial outer membrane or peroxisomal membrane, and a proline-rich region that mediates the interactions with its downstream signaling partners tumor necrosis factor receptor-related factor 2 (TRAF2), TRAF3, TRAF5, and TRAF6 [19,20]. The signal is then relayed by the kinases TBK1 and IKKε to the transcription factors interferon regulatory factor 3 (IRF3), IRF7, and nuclear factor-κB (NF-κB). These translocate to the nucleus and activate the transcription of genes encoding type I interferons (IFN-I) and proinflammatory cytokines. Subsequently, IFN-I signals through the IFN-α/β receptor (IFNAR) to activate the transcription of hundreds of interferon-stimulated genes (ISGs), which then mount a potent antiviral response [21,22]. Post-translational modifications, such as phosphorylation and ubiquitination, play a crucial role in the activation and downregulation of this pathway [19,23].

Viruses have developed a vast array of mechanisms to evade antiviral responses. This includes shielding their genome from recognition by nucleic acid sensors, inhibition of the protein–protein interactions that are required for signal transduction, and modulation of the post-translational modifications that control the activation of the innate immune response pathways [24,25]. Furthermore, viral proteases, which play a vital role in viral polyprotein processing, can also cleave cellular proteins, such as the RLRs and MAVS [11,26]. For example, RIG-I can be cleaved by SARS-CoV-2 M^pro^ [13]. Picornaviruses encode a 2A protease (2Apro) and a 3C protease (3Cpro) that, similar to the coronavirus M^pro^, adopt a chymotrypsin-like fold [11]. The 3Cpro of various picornavirus species has been shown to cleave RIG-I [27,28], while the 3Cpro of Theilovirus, another member of the family *Picornaviridae*, cleaves MDA5 [29]. In addition to cleavage of RIG-I and MDA5, cleavage of MAVS has also been reported for several picornaviruses [30,31,32,33,34]. Both the 2Apro and the 3Cpro of picornaviruses seem to be able to cleave MAVS. Furthermore, the NS3/4A protease of the flaviviruses hepatitis C virus and GB virus B and M^pro^ of the arterivirus porcine respiratory and reproductive syndrome virus (PRRSV) can also cleave MAVS [35,36,37]. M^pro^ of coronaviruses and arteriviruses, which both belong to the order *Nidovirales*, and the 3Cpro of picornaviruses also cleave another protein, NF-κB essential modulator (NEMO), which controls the activation of NF-κB [38,39,40,41,42,43,44]. Although cleavage of the indicated targets is often partial, all these cleavage events together contribute to the suppression of the antiviral immune response.

Here, we show for the first time that MAVS is cleaved during Middle East respiratory syndrome (MERS)-CoV infection and that the expression of MERS-CoV M^pro^ can induce cleavage of MAVS upon overexpression. Furthermore, M^pro^ expression reduced the IFN-β and NF-κB response, which could be the result of MAVS cleavage combined with various other immune evasive mechanisms.

## 2. Materials and Methods

### 2.1. Cell Culture

Huh7 cells (a kind gift from Dr. Ralf Bartenschlager, Heidelberg University, Germany) were grown in Dulbecco’s modified Eagle’s medium (DMEM, Gibco) supplemented with 8% fetal calf serum (FCS, Bodinco BV, Alkmaar, The Netherlands), 0.1 mM non-essential amino acids (NEAA, Lonza, Verviers, France), 2 mM L-glutamine, 50 IU/mL penicillin, and 50 µg/mL streptomycin (all Sigma-Aldrich). HEK293T cells (ATCC, #CRL-3126, Gaithersburg, Maryland, USA) were cultured in DMEM supplemented with 10% FCS, 2 mM L-glutamine, 50 IU/mL penicillin, and 50 µg/mL streptomycin. Both cell lines were cultured at 37 °C in 5% CO_2_.

### 2.2. Virus Infections

Infection of Huh7 cells with MERS-CoV (EMC/2012 strain, accession number NC_019843) or Huh7-adapted SARS-CoV-2 (a kind gift from Dr. Dirk Jochmans and Dr. Johan Neyts, KU Leuven, Belgium, [45]) was performed in infection medium consisting of minimum essential medium with Earle’s salts (EMEM, Gibco, Thermo Fisher Scientific, Landsmeer, The Netherlands), supplemented with 2% FCS, 0.1 mM NEAA, 2 mM L-glutamine, 50 IU/mL penicillin, and 50 µg/mL streptomycin. The cells were inoculated for 1 h, after which they were washed three times with PBS and the infection medium was added. To investigate the involvement of various cellular pathways in MAVS cleavage, 20–25 µM Z-VAD(OMe)-FMK (MedChemExpress, Monmouth Junction, New Jersey, USA) or 20 nM wortmannin (MedChemExpress) was added immediately after removing the inoculum, while 20 µM MG132 (Sigma) was added 4 h prior to harvesting the samples. Z-VAD efficacy was assessed using a CyQUANT™ LDH cytotoxicity assay (Thermo Fisher Scientific), according to the manufacturer’s instructions, while the efficacy of wortmannin and MG132 was assessed using western blot analysis of the autophagy marker LC3B-II and conjugated ubiquitin levels, respectively. Virus titers were determined using plaque assays on Huh7 cells for 2–3 days at 37 °C, as described previously [46]. All experiments involving live MERS-CoV or SARS-CoV-2 were performed in a biosafety level 3 (BSL-3) laboratory.

### 2.3. Plasmids

The following plasmids have been described previously: pLuc-IFN-β [47], pcDNA3.1-PL^pro^ (MERS-CoV)-V5 wild-type and C1592A [48], and pcDNA3.1-FLAG-MAVS [32]. Expression plasmids for MAVS mutants were generated using site-directed mutagenesis. pLuc-NF-κB was a kind gift from Dr. Adolfo Garcia-Sastre (Icahn School of Medicine at Mount Sinai, New York, United States of America). pGL3-ISRE, expressing firefly luciferase under the control of the ISG54 promoter that contains interferon-sensitive response elements (ISRE), was a kind gift from Dr. Gijs Versteeg (Max Perutz Labs, Vienna, Austria). pRL-TK, which expresses Renilla luciferase under the control of the constitutively active HSV thymidine kinase promoter, was purchased from Promega. To generate plasmids for the expression of M^pro^ in mammalian cells, the mammalian codon-optimized sequences encoding MERS-CoV M^pro^ (pp1a amino acids 3242–3553), SARS-CoV M^pro^ (pp1a amino acids 3241–3546), and SARS-CoV-2 M^pro^ (pp1a amino acids 3264–3569) were cloned into a pcDNA3.1 vector in frame with an N-terminal (SARS-CoV and SARS-CoV-2) or C-terminal (MERS-CoV) V5 tag. To prevent cleavage of the C-terminal V5 tag by MERS-CoV M^pro^, residue Q3553 was mutated to proline using site-directed mutagenesis. 

### 2.4. Transfections and Analysis of Protein-Protein Interaction

To assess whether MAVS can be cleaved by one of the coronavirus proteases, HEK293T cells were transfected with 1 µg pcDNA3.1-FLAG-MAVS and 1 µg pcDNA3.1 encoding various proteases, namely MERS-CoV M^pro^ wild-type or C148A, MERS-CoV PL^pro^ wild-type or C1592A, SARS-CoV wild-type or C145A, or SARS-CoV-2 wild-type or C145A. To assess the dose-dependent cleavage of MAVS by MERS-CoV M^pro^, cells were transfected with 1 µg pcDNA3.1-FLAG-MAVS and 100, 250, 500, 750, or 1000 ng pcDNA3.1-M^pro^-V5 wild-type or 1000 ng pcDNA3.1-M^pro^-V5 C148A. The DNA was transfected using linear polyethylenimine (PEI 25K, Polysciences, Eppelheim, Germany) in a 1:3 DNA-to-PEI ratio. At 24 h post-transfection, cell lysates were harvested in 2x Laemmli sample buffer (2x LSB, 250 mM Tris pH 6.8, 4% SDS, 20% glycerol, 10 mM DTT, 0.01% bromophenol blue).

The interaction between MAVS and M^pro^ was assessed using immunoprecipitation. To this end, 60–80% confluent HEK293T cells in 10 cm dishes were transfected with 7.5 µg pcDNA3.1-FLAG-MAVS and 7.5 µg pcDNA3.1-M^pro^-V5 wild-type or C148A. At 24 h post-transfection, the cells were lysed in 20 mM Tris-HCl pH 7.4, 135 mM NaCl, 1% Triton X-100, 10% glycerol, and cOmplete protease inhibitor (Merck Millipore, Darmstadt, Germany). Subsequently, the lysates were incubated with anti-FLAG beads (Pierce™ Anti-DYKDDDDK Magnetic Agarose, Thermo Fisher Scientific) to immunoprecipitate FLAG-MAVS. After 4 h incubation at room temperature, the samples were eluted in 2x LSB, followed by boiling for 10 min. The samples were then analyzed by western blot for the presence of M^pro^.

### 2.5. IFN-β, ISRE, and NF-κB Luciferase Assays

The activity of the IFN-β, ISRE, and NF-κB promoters was assessed using a luciferase reporter assay. For the IFN-β and ISRE luciferase assays, 60–80% confluent HEK293T cells in 24-well plates were transfected with plasmids encoding pLuc-IFN-β or pGL3-ISRE (50 ng), pRL-TK (5 ng), MAVS (25 ng), and MERS-CoV M^pro^ (25–200 ng). For the NF-κB luciferase assay, 60–80% confluent HEK293T cells were transfected with plasmids encoding pLuc-NF-κB (50 ng), pRL-TK (5 ng), MAVS (50 ng), and MERS-CoV M^pro^ wild-type (25–200 ng) or C148A (200 ng). pcDNA3.1(-) was added to a total of 800 ng DNA per well (24-well format). The DNA was mixed with PEI in a 1:3 DNA-to-PEI ratio. Transfections were performed in quadruplicate. At 24 h post-transfection, the luciferase activity was measured using the dual-luciferase reporter assay system (Promega, Leiden, The Netherlands). The firefly luciferase values were normalized to Renilla luciferase values to correct for variations in the transfection efficiency. All values are shown relative to the cells that were transfected with MAVS only. The statistical analyses were conducted using one-way ANOVA with Dunnett’s multiple comparisons test, comparing each group to the MAVS-induced control. The cell lysates were mixed with 2x LSB in a 1:1 ratio for the western blot analysis of M^pro^ expression.

### 2.6. Western Blot Analysis

Whole-cell lysates were separated using sodium dodecyl sulfate–polyacrylamide gel electrophoresis (SDS-PAGE) and transferred onto a 0.2 µm PVDF membrane (Amersham Hybond-LFP 0.2 µm PVDF, Sigma, Zwijndrecht, The Netherlands). The membranes were blocked for 1 h at room temperature in 1% casein in PBS with 0.05% Tween-20 (PBST). Primary antibody incubation was conducted overnight at 4 °C in 1% casein in PBST. The following primary antibodies were used: mouse monoclonal anti-MAVS (clone E-3, Santa Cruz, used at 1:200 for endogenous MAVS and 1:500 for overexpressed MAVS), mouse monoclonal anti-β-actin (clone AC74, Sigma, 1:5000), mouse monoclonal anti-V5 (clone 2F11F7, Thermo Fisher Scientific, 1:5000), mouse monoclonal anti-FLAG (clone M2, Merck Millipore, 1:5000), mouse monoclonal anti-LC3B (M186-3, MBL, 1:1000), mouse monoclonal anti-ubiquitin (clone A-5, Santa Cruz, Heidelberg, Germany, 1:500), mouse monoclonal anti-α-tubulin (clone B-5-1-2, Sigma, 1:5000), and polyclonal rabbit anti-membrane (M) protein of SARS-CoV (cross-reacts with SARS-CoV-2 M protein, 1:1000 [49]). MERS-CoV membrane (M) protein was detected using a polyclonal rabbit serum (clone R9004, 1:1000) that was raised against a C-terminal peptide in the M protein (CRYKAGNYRSPPITADIELALLRA). After primary antibody incubation, the membranes were incubated with horseradish peroxidase (HRP)-conjugated secondary antibodies against mouse IgG (ab205719, Abcam, Cambridge, United Kingdom) or rabbit IgG (P0217, Agilent Dako, Amstelveen, The Netherlands) or with biotin-conjugated anti-mouse IgG (31800, Thermo Fisher Scientific) or anti-rabbit IgG (A16033, Thermo Fisher Scientific). In the case of biotin-conjugated secondary antibodies, the membranes were subsequently incubated with Cy3-conjugated mouse-anti-biotin (200-162-211, Jackson ImmunoResearch, Newmarket, United Kingdom). The protein bands were detected using Clarity Western ECL substrate (BioRad, Veenendaal, The Netherlands) or Cy3 fluorescent signals, and visualized using an Alliance Q9 Advanced imaging system (Uvitec, Cambridge, United Kingdom).

### 2.7. Immunofluorescence Microscopy

Infected cells were fixed with 3% paraformaldehyde in PBS for at least 4 h. The cells were permeabilized with 0.1% Triton-X100 in PBS and stained with primary antibodies in 5% FCS in PBS for 1 h. The primary antibodies used were anti-MAVS (clone E-3, Santa Cruz, 1:50 dilution), anti-TOMM20 (ab78547, Abcam, 1:200), and anti-NSP5 of SARS-CoV for the detection of M^pro^ (cross-reacts with MERS-CoV NSP5, 1:400 [49]). As secondary antibodies, Cy3-conjugated donkey-anti-mouse IgG (711-165-152, Jackson ImmunoResearch, 1:1000) and Alexa488-conjugated goat-anti-rabbit IgG (A-11001, Thermo Fisher Scientific, 1:300) were used. The nuclei were stained with Hoechst 33342 (Thermo Fisher Scientific). Coverslips were mounted using ProLong glass antifade mounting fluid (Thermo Fisher Scientific). Images were acquired using a Leica DM6B fluorescence microscope (Leica Microsystems B.V., Wetzlar, Germany).

## 3. Results

### 3.1. MAVS Is Cleaved in MERS-CoV-Infected Cells

Since MAVS is a crucial mediator in the immune response against MERS-CoV infection, we analyzed MAVS protein levels in infected cells by western blot. To this end, we performed MERS-CoV infection of Huh7 cells, which support MERS-CoV replication well and are immune competent [50,51]. As the infection in Huh7 cells progressed, the abundance of full-length MAVS decreased, which coincided with the appearance of two additional bands of about 20 and 35 kDa that were recognized by a MAVS-specific antibody (Figure 1A,B). This strongly suggests that MAVS is cleaved during MERS-CoV infection. Since the picornavirus 3Cpro and PRRSV M^pro^ have been shown to be able to cleave MAVS, we wondered whether MERS-CoV M^pro^ plays a role in the cleavage of MAVS. To get a first indication, we analyzed the localization of MAVS and M^pro^ upon MERS-CoV infection. We found partial overlap in the staining patterns for M^pro^ and MAVS (Figure 1C). Interestingly, the staining pattern for MAVS was quite different between infected and mock-infected cells, with the pattern being more punctuated and dispersed in infected cells. This made us wonder whether MAVS changes localization upon infection. We then stained both mock-infected and infected cells for MAVS and the mitochondrial marker TOMM20 and found that MAVS still localized to the mitochondria, but the morphology of the mitochondria had changed upon infection (Figure 1D). In some infected cells, the mitochondria were more elongated and seemed to be fused, while in other cells the mitochondria became more dispersed. Together, these results demonstrate that MAVS is cleaved during infection and the close proximity of MAVS and M^pro^ could leave MAVS susceptible to M^pro^ cleavage.

### 3.2. MERS-CoV M^pro^ Induces MAVS Cleavage upon Overexpression

The MERS-CoV genome encodes two proteases, PL^pro^ and M^pro^. To test whether either of these proteases triggers MAVS cleavage, a plasmid encoding N-terminally FLAG-tagged MAVS was expressed in HEK293T cells together with plasmids encoding wild- type or catalytically inactive PL^pro^ or M^pro^. Western blot analysis showed that expression of M^pro^, but not PL^pro^, induced MAVS cleavage in a dose-dependent manner (Figure 2A,B). Furthermore, cleavage was dependent on the protease activity of M^pro^, since a catalytically inactive mutant did not induce MAVS cleavage. We observed fragments at 20 and 35 kDa that were similar to those observed in infected cells (Figure 1A), although the abundance of the 35 kDa fragment was much lower than that of the 20 kDa fragment upon overexpression. Additionally, we observed a fragment at 55 kDa. This is most likely generated by caspase-mediated cleavage of MAVS, since multiple reports show cleavage of MAVS at 55 kDa by apoptotic caspases [52,53,54]. Overexpression of wild-type M^pro^ did induce mild cytopathic effects (CPE). To probe whether there was a direct interaction between MAVS and M^pro^, we precipitated MAVS using anti-FLAG beads. Both wild-type and catalytically inactive M^pro^ were detected in the precipitate, showing that they interact with MAVS (Figure 2C).

We then set out to identify the cleavage sites at which M^pro^ could cleave MAVS. Therefore, we immunoprecipitated the MAVS fragments using FLAG beads but unfortunately, we could not obtain sufficient quantities for analysis by mass spectrometry. The consensus cleavage site of MERS-CoV M^pro^ is defined as (V/P/M/L)Q↓(S/A/G/N) [55]. Based on this profile and the size of the fragments, we identified five glutamine residues whose position in the MAVS sequence could match the fragments that we observed (Figure 2D). We created various MAVS mutants, in which one of these glutamines was mutated to alanine and one mutant in which all five glutamines were mutated to alanine. We then expressed these mutants together with M^pro^ and analyzed whether they were still cleaved. Mutation of Q145, Q148, Q159, Q162, and Q196 to alanine, either alone or in combination, did not prevent M^pro^ from inducing MAVS cleavage (Figure 2E). These results suggest that M^pro^ interacts with and specifically cleaves MAVS via its proteolytic activity upon overexpression, however the cleavage site remains to be elucidated.

### 3.3. MAVS Cleavage in MERS-CoV-Infected Cells Is Independent of Apoptosis, Proteasome-Mediated Degradation, and Autophagy

Although the data thus far indicate that M^pro^ is likely responsible for the cleavage of MAVS during infection, we also investigated some cellular processes that could lead to the cleavage or degradation of MAVS. It has been reported that MAVS can be cleaved by apoptotic caspases at residues D429 and D490, leading to the generation of a fragment of 45–55 kDa [52,53,54]. Although the MAVS fragments that are generated by caspase cleavage are of a different molecular weight than the fragments that we observed, we wanted to exclude that the cleavage of MAVS upon MERS-CoV infection is mediated by caspases. Therefore, we treated infected cells with the pan-caspase inhibitor Z-VAD. Western blot analysis showed that the cleavage of MAVS is not affected by Z-VAD, thus showing that it is not mediated by caspases (Figure 3A). Z-VAD can reduce lytic cell death in the context of coronavirus infection [56]. To confirm that Z-VAD effectively prevented caspase activation and subsequent lytic cell death, cytotoxicity was measured using an LDH release assay. Z-VAD effectively reduced LDH release in the infected cells to similar levels as the mock-infected control (Figure 3B).

To assess whether proteasome-mediated degradation played a role in MAVS cleavage, infected cells were treated with the proteasome inhibitor MG132 from 20 to 24 h post-infection, after which samples were collected. The accumulation of ubiquitin conjugates on western blot showed that MG132 treatment was effective, but cleavage of MAVS still occurred (Figure 3C). Furthermore, we inhibited autophagy using wortmannin, but wortmannin treatment also did not prevent MAVS cleavage, while the detection of LC3B, a marker for autophagic flux, showed that wortmannin treatment was effective (Figure 3D). It is thus highly likely that the cleavage of MAVS occurs independent of cellular degradation pathways and is initiated by the virus.

### 3.4. M^pro^ Inhibits MAVS-Induced Immune Responses

It has previously been reported that cleavage of MAVS by viral proteases is a means of antagonizing IFN-I and NF-κB signaling [29,30,35,37]. Therefore, we assessed whether M^pro^ expression interfered with the ability of MAVS to mount an immune response. To this end, HEK293T cells were co-transfected with expression vectors for MAVS, wild-type or catalytically inactive M^pro^, and plasmids encoding firefly luciferase under the control of the IFNβ, ISRE, or NF-κB promoter, as well as Renilla luciferase expressed from a constitutively active promoter to correct for differences in transfection efficiency. IFN-β, ISRE, and NF-κB promoter activities were all reduced by wild-type M^pro^ in a dose-dependent manner (Figure 4). ISRE promoter activity was modestly increased in the presence of catalytically inactive M^pro^ (Figure 4B), while NF-κB promoter activity was slightly reduced (Figure 4C). Overall, the expression of wild-type M^pro^ inhibited MAVS-mediated activation of the innate immune response, which could at least in part be the result of cleavage of MAVS.

### 3.5. MAVS Is Not Cleaved upon Infection with SARS-CoV-2

We wondered whether other coronaviruses could also induce cleavage of MAVS. Therefore, we infected Huh7 cells with MERS-CoV or Huh7-adapted SARS-CoV-2. While MERS-CoV infection led to the cleavage of MAVS, MAVS was not cleaved upon infection with SARS-CoV-2 (Figure 5A), even though virus replication and release of viral progeny at 24 h post-infection was comparable to that of MERS-CoV (Figure 5B). In addition, we assessed whether SARS-CoV or SARS-CoV-2 M^pro^ could cleave MAVS upon co-expression in HEK293T cells. In line with the lack of MAVS cleavage in SARS-CoV-2-infected cells, we did not observe cleavage of MAVS by either SARS-CoV or SARS-CoV-2 M^pro^ (Figure 5C). Overall, this suggests that cleavage of MAVS specifically occurs upon infection with MERS-CoV.

## 4. Discussion

MAVS bridges the recognition of viral dsRNA by RIG-I and MDA5 with the activation of IRF3 and NF-κB and is therefore an important signaling adaptor in the innate immune response against RNA viruses. It is thus not surprising that several viruses target MAVS to facilitate their replication. In this study, we show for the first time that MAVS is cleaved in MERS-CoV-infected cells, and that this cleavage is likely induced by the viral 3C-like protease M^pro^. Interestingly, cleavage did not occur in SARS-CoV-2-infected cells, nor could SARS-CoV or SARS-CoV-2 M^pro^ induce MAVS cleavage. This is in line with a previous report that showed that SARS-CoV-2 M^pro^ could cleave RIG-I, but not MAVS [13]. Although the 3D structures of MERS-CoV M^pro^ and SARS-CoV-2 M^pro^ are very similar, they share only 50% sequence identity [57]. Therefore, it could be that residues that are required for the interaction and subsequent cleavage of MAVS by MERS-CoV M^pro^ are not present in SARS-CoV-2 M^pro^. Although we did not find cleavage of MAVS in SARS-CoV-2-infected cells, SARS-CoV-2 can manipulate MAVS signaling in other ways. The SARS-CoV-2 nucleocapsid protein (N) was shown to interact with MAVS and reduced its activation by direct interaction, as well as by modifying post-translational modifications that are required for MAVS activation and downstream signaling [58,59,60]. It is not known whether the MERS-CoV N protein could also interact with MAVS. In addition, the SARS-CoV-2 accessory protein ORF3c was shown to interact with MAVS and to induce cleavage of MAVS by caspases, altogether leading to a reduction in IFN-β promoter activity [61]. ORF3c is only expressed by viruses from the subgenus *Sarbecovirus* and is not expressed by MERS-CoV [62]. This underscores that different coronaviruses can evade the innate immune response using diverse and complementary mechanisms.

Cleavage of MAVS by both the 2A and the 3C proteases has been reported for several picornaviruses [30,31,32,33,34]. In contrast to the picornavirus proteases that both cleave MAVS at a single position, we did not find one but two fragments of MAVS, both of which can be detected using an antibody against the N-terminal domain of MAVS. This suggests that, in the case of MERS-CoV, MAVS is cleaved at multiple positions. The 3Cpro of picornaviruses was shown to cleave MAVS after Q148 [30,32]. Therefore, we tested whether M^pro^ would also cleave MAVS after Q148. However, when Q148 was mutated to alanine, cleavage still occurred, suggesting that M^pro^ cleaves at a different site. Next, we also mutated the other glutamines whose position in the MAVS sequence could match the fragments that we observed. However, M^pro^ was still able to induce cleavage of all of these MAVS mutants, even when all the glutamine-to-alanine mutations were combined in one MAVS mutant (Figure 2C). Interestingly, recent reports that characterize the consensus cleavage site motifs of the M^pro^ proteins of SARS-CoV-2 and hCoV-NL63, suggest that in some cases M^pro^ can also accommodate a histidine or methionine at P1, although glutamine is still preferred [63,64]. This implies that MAVS cleavage by M^pro^ does not have to occur after a glutamine, but could potentially also happen at other positions, which requires further investigation of the fragments using mass spectrometry. However, for this, one needs to be able to purify the fragments in sufficient quantities. We did find an interaction between MAVS and M^pro^ in co-immunoprecipitation experiments (Figure 2D). Furthermore, cleavage of MAVS was dependent on the catalytic activity of M^pro^ (Figure 2A). It is thus likely that M^pro^ associates with MAVS to mediate its cleavage, but we cannot exclude the involvement of a cellular protein.

During coronavirus infection, many viral proteins cooperate to evade the immune response [6]. Although the main function of M^pro^ is to release viral non-structural proteins from the replicase polyproteins, there are reports on different coronaviruses showing that M^pro^ can also contribute to immune evasion by targeting various proteins that mediate the innate immune response [41,43,44,65]. Here, we show that the catalytic activity of MERS-CoV M^pro^ reduces MAVS-induced activation of the IFN-β, ISG, and NF-κB responses (Figure 5), which could in part be the result of M^pro^-mediated cleavage of MAVS. In addition to M^pro^, several other MERS-CoV proteins can interfere with the RLR pathway, such as the PL^pro^ domain in NSP3, the endonuclease domain in NSP15, and the accessory proteins NS4a and NS4b [48,66]. However, M^pro^ is the only MERS-CoV protein that has been shown to target MAVS specifically. PL^pro^ most likely removes ubiquitin from various proteins in the RLR pathway, thereby preventing its activation. NSP15, NS4a, and NS4b are thought to prevent the accumulation of dsRNA, thereby removing the signal that activates the RLR pathway [47,58]. Altogether, these mechanisms complement each other, and their combined activity leads to effective suppression of the innate immune response.

During infection, MAVS is not only targeted for degradation by the virus, but there are also cellular mechanisms in place that keep MAVS activation at bay to prevent excessive activation of the immune response. MAVS is subject to negative regulatory mechanisms, such as K48-linked ubiquitination leading to its proteasome-mediated degradation, autophagy-mediated degradation, and caspase-mediated inactivation [52,67,68,69]. To distinguish whether the cleavage of MAVS during infection was triggered by the virus or was part of a cellular negative feedback mechanism, we inhibited the proteasome and the autophagy machinery during infection. MAVS cleavage was still observed under these conditions, suggesting that it is neither the result of proteasome-mediated degradation nor of autophagy-mediated degradation (Figure 3C,D). In addition, we showed that MAVS cleavage was not dependent on the activity of caspases (Figure 3A). It is thus highly likely that cleavage of MAVS is not the result of cellular negative regulation of the immune response but is induced by the virus as an immune evasive mechanism. Ideally, one would inhibit M^pro^ during infection to formally prove that M^pro^ is responsible for cleaving MAVS. Several inhibitors have been developed and one of them, nirmatrelvir, is authorized for clinical use in coronavirus-induced disease 19 (COVID-19) patients [70,71,72]. However, using these inhibitors for mechanistic studies into the role of M^pro^ in immune evasion during infection is complicated because of the crucial role of M^pro^ in the processing of the viral replicase polyproteins. Inhibition of M^pro^ leads to a strong reduction in virus replication [73]. Since the immune response is directly correlated with the viral load, it is hard to distinguish whether changes upon inhibition of M^pro^ are a direct effect of M^pro^ or a more general effect of the inhibition of viral replication. Further studies are thus required to understand the mechanisms by which M^pro^ induces MAVS cleavage, so that uncleavable mutants of MAVS can be generated to understand their effect on the immune response against MERS-CoV. Additionally, it would be very interesting to study whether the MAVS fragments can interfere with its activation and downstream signaling, as it was shown that shorter isoforms of MAVS act as negative regulators by preventing MAVS oligomerization and by competing for the downstream signaling partners TRAF2 and TRAF6 [74,75,76]. Future studies should investigate the effect of M^pro^-mediated cleavage of MAVS on the activation and oligomerization of MAVS.

Altogether, we found that MAVS is cleaved in Huh7 cells infected with MERS-CoV and that MERS-CoV M^pro^ could mediate cleavage of MAVS upon overexpression. Although we could exclude the involvement of several cellular pathways, the exact mechanism by which cleavage occurs in infected cells needs to be investigated further. Cleavage of host substrates, particularly MAVS in this context, is one of the many viral mechanisms contributing to suppression of the immune response, thereby enhancing viral pathogenesis. Cleavage of MAVS was already described for flaviviruses and picornaviruses, but we now show it also for coronaviruses, although the mechanism is not shared by all coronaviruses. These insights highlight an interesting difference in immune evasion between highly pathogenic coronaviruses. Furthermore, they underscore the potential of M^pro^ inhibitors not only in directly inhibiting virus replication, but also in blocking a part of the immune evasive mechanisms of the virus, thus enhancing the effectiveness of these inhibitors even further.

## Figures and Tables

**Figure 1 viruses-16-00256-f001:**
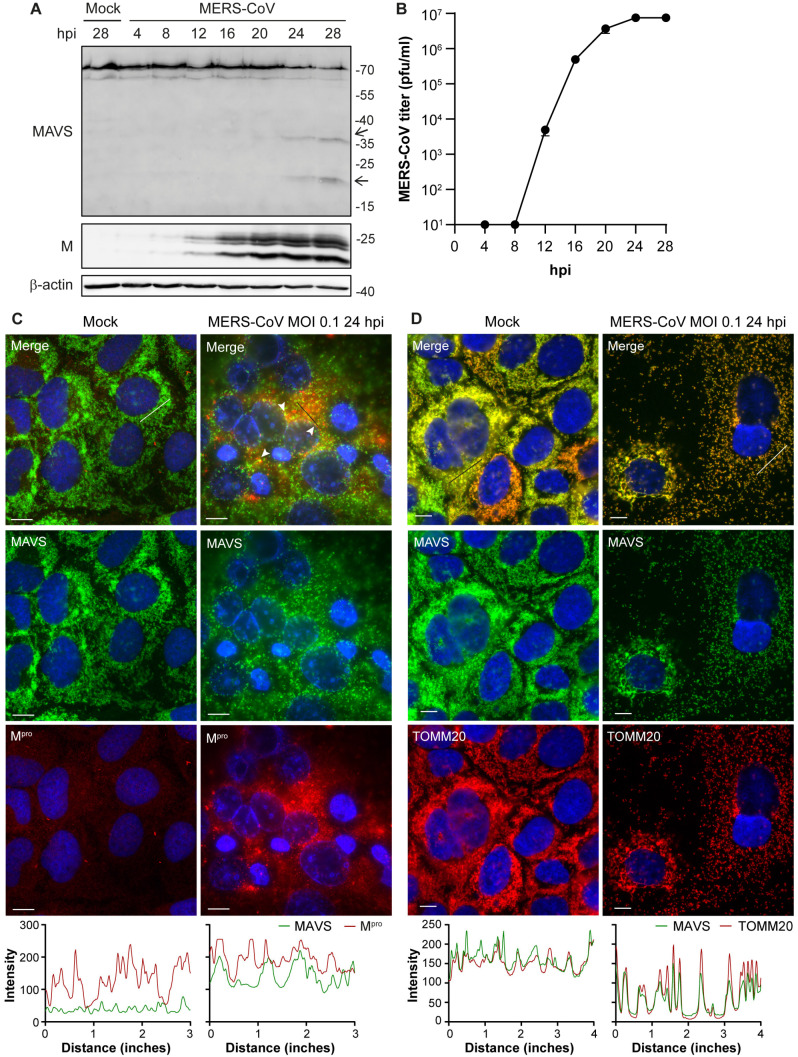
MAVS is cleaved in Huh7 cells infected with MERS-CoV. Huh7 cells were infected with MERS-CoV at MOI 0.1 and samples were collected at various timepoints. (**A**) MAVS and viral membrane (M) protein levels were analyzed by western blot. Arrows indicate MAVS cleavage products. (**B**) Virus titers in the supernatants were determined by plaque assay. (**C**,**D**) Immunofluorescence microscopy analysis of the localization of MAVS (**C**,**D**), M^pro^ (**C**), and TOMM20 (**D**) in mock- and MERS-CoV-infected Huh7 cells at 24 hpi. Line profiles (black/white line) depict intensity of the fluorescent signal for MAVS (**C**,**D**), M^pro^ (**C**), and TOMM20 (**D**). Arrowheads indicate areas of colocalization. Scale bars in the bottom left corner of each image are 10 µm. Representative examples of three biological replicates are shown.

**Figure 2 viruses-16-00256-f002:**
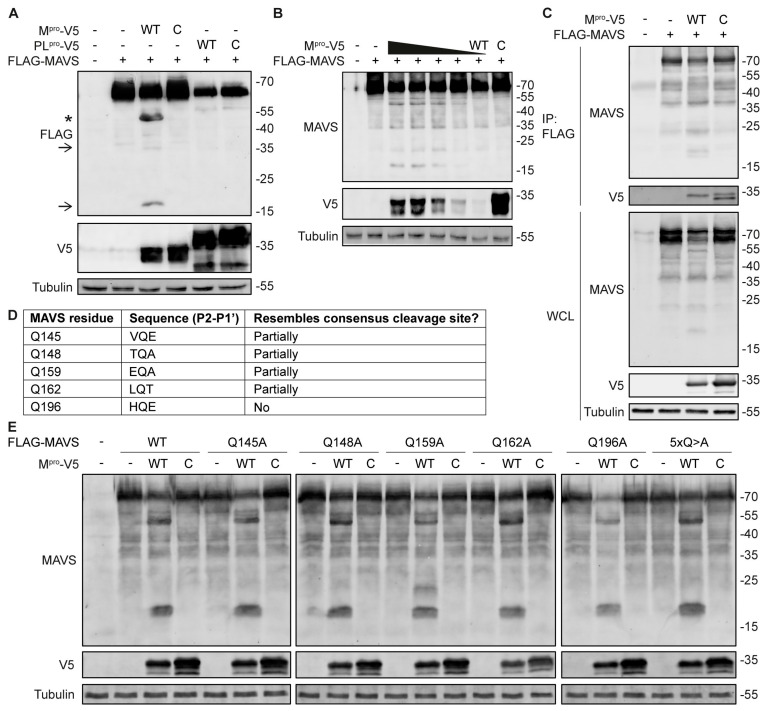
MERS-CoV M^pro^ interacts with MAVS and induces its cleavage. (**A**) HEK293T cells were transfected with plasmids expressing MAVS and wild-type (WT) or catalytically inactive (C) M^pro^ or PL^pro^ of MERS-CoV. Cleavage of MAVS was analyzed by western blot. Arrows indicate MAVS fragments. * Indicates the MAVS fragment that corresponds to the caspase-generated fragment reported in the literature. (**B**) HEK293T cells were transfected with plasmids encoding MAVS and various amounts of MERS-CoV M^pro^, and MAVS cleavage was analyzed by western blot. (**C**) Co-immunoprecipitation analysis of MAVS and wild-type or catalytically inactive M^pro^. (**D**) Table containing the glutamine residues in MAVS that were mutated to identify the cleavage site, their sequence background, and their resemblance to the consensus cleavage site of MERS-CoV M^pro^, which is defined as (V/P/M/L)Q↓(S/A/G/N). (**E**) HEK293T cells were transfected with plasmids encoding MERS-CoV M^pro^ and various glutamine-to-alanine mutants of MAVS, and cleavage of MAVS was analyzed by western blot. 5xQA indicates the combination of Q145A, Q148A, Q159A, Q162A, and Q196A. Representative examples of at least three biological replicates are shown.

**Figure 3 viruses-16-00256-f003:**
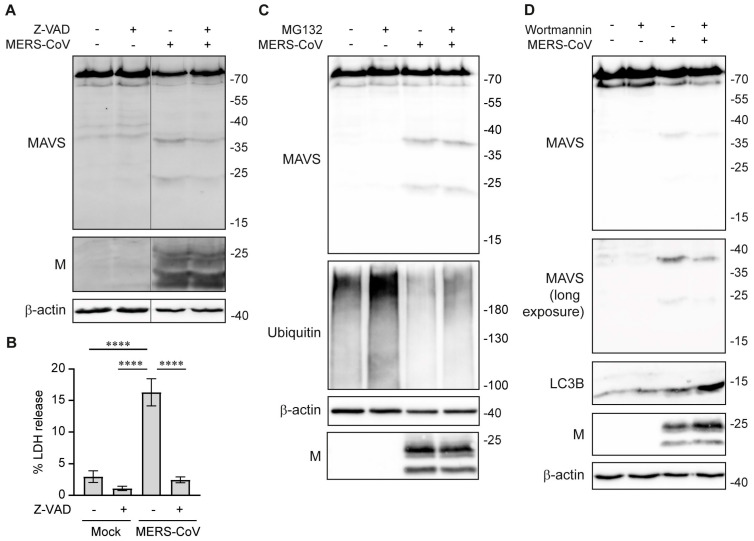
Cleavage of MAVS is independent of caspases, proteasome-mediated degradation, and autophagy. (**A**) Huh7 cells were infected with MERS-CoV at MOI 0.1 and treated with 20 µM Z-VAD. Cleavage of MAVS at 24 hpi was analyzed by western blot. (**B**) Cytotoxicity upon infection and treatment with Z-VAD was determined using LDH release assay on cell culture supernatants. (**C**) Huh7 cells were infected with MERS-CoV at MOI 0.1 and treated with 20 µM MG132 during the last 4 h of the infection. Cleavage of MAVS at 24 hpi was analyzed by western blot. (**D**) Huh7 cells were infected with MERS-CoV at MOI 0.1 and treated with 20 nM wortmannin. Cleavage of MAVS at 22 hpi was analyzed by western blot. Representative examples are shown of at least three biological replicates. Data in panel **B** are represented as mean ± SD of 3 experiments and were analyzed by one-way ANOVA with Dunnett’s multiple comparisons test, comparing each group to the MAVS control. **** *p* < 0.0001.

**Figure 4 viruses-16-00256-f004:**
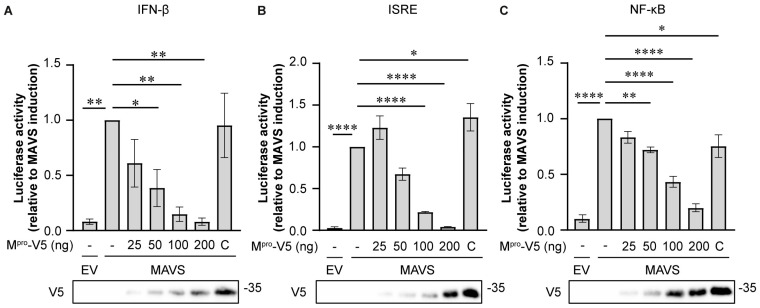
M^pro^ inhibits the innate immune response. HEK293T cells were transfected with firefly luciferase under the control of the IFN-β (**A**), ISRE (**B**), or NF-κB (**C**) promoter, in combination with MAVS or empty vector (EV) and various amounts of MERS-CoV M^pro^. Luciferase activity was determined at 24 h post-transfection and normalized to the sample that was only transfected with MAVS. Data are represented as mean ± s.e.m. of 3 experiments and were analyzed by one-way ANOVA with Dunnett’s multiple comparisons test, comparing each group to the MAVS control. * *p* < 0.05, ** *p* < 0.01, **** *p* < 0.0001.

**Figure 5 viruses-16-00256-f005:**
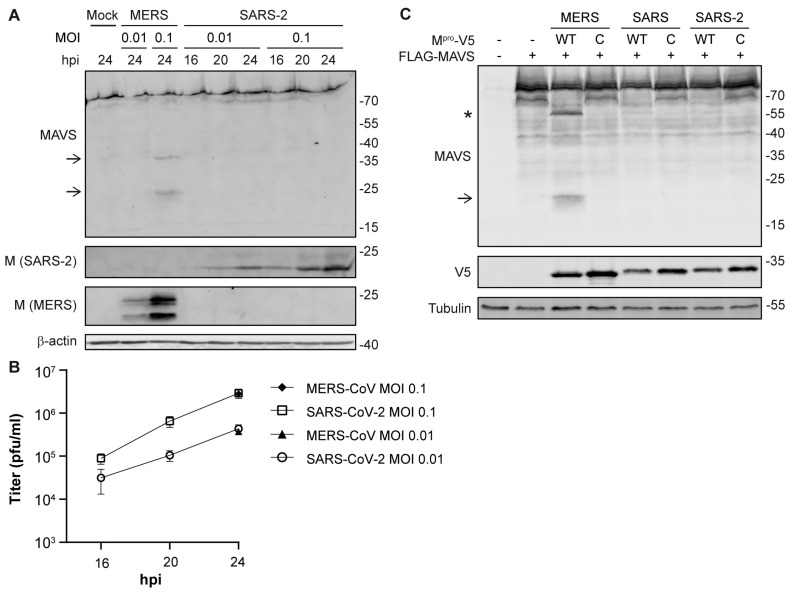
MAVS cleavage is not induced by other coronaviruses. Huh7 cells were infected with MERS-CoV or SARS-CoV-2 at MOI 0.01 or 0.1 and samples were collected at various timepoints. (**A**) MAVS and viral M protein levels were analyzed by western blot. (**B**) Virus titers in the supernatants were determined by plaque assay. Arrows indicate MAVS fragments. Asterisk indicates the MAVS fragment that corresponds to the caspase-generated fragment reported in the literature. (**C**) HEK293T cells were transfected with plasmids encoding MAVS and wild-type or catalytically inactive (C) M^pro^ of MERS-CoV, SARS-CoV, or SARS-CoV-2. Cleavage of MAVS was analyzed by western blot. Representative examples of three biological replicates are shown.

## Data Availability

All data supporting the reported results can be found in the published paper.

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
