# Peer review of "The Main Protease of Middle East Respiratory Syndrome Coronavirus Induces Cleavage of Mitochondrial Antiviral Signaling Protein to Antagonize the Innate Immune Response"

_viruses, 2024, doi:10.3390/v16020256_

Round 1
Reviewer 1 Report
Comments and Suggestions for Authors
In this manuscript titled “The main protease of Middle East respiratory syndrome coronavirus induces cleavage of MAVS to antagonize the innate immune response”, the authors show that MAVS is cleaved in cells infected with Middle East respiratory syndrome coronavirus (MERS-CoV), but not in cells infected with severe acute respiratory syndrome coronavirus-2 (SARS-CoV-2). The results confirmed a novel mechanism by which MERS-CoV downregulates the innate immune response, which is not observed among other highly pathogenic coronaviruses. However, there exist many minor problems in this manuscript, which need further revision and improvement. The specific amendments are as follows:
1. The main question addressed by the research is that the authors discover a novel mechanism by which MERS-CoV downregulates the innate immune response, which is not observed among other highly pathogenic coronaviruses.
2. This topic is relevant to the field and addresses a specific gap in the field.
3. The authors discover a novel mechanism by which MERS-CoV downregulates the innate immune response.
4. The methodology in the manuscript needs no improvement.
5. The conclusion is consistent with the proposed evidence and arguments, and solves the main problems raised.
6. The references are appropriate.
7. It is suggested to improve the pictures in the manuscript.
8. In this manuscript, “IFNβ” and “NFκB” should be written as “IFN-β” and “NF-κB”.
9. In line 36, “nsps” should be written as “NSPs”.
10. In line 95, “CO2” should be written as “CO2”.
11. The SDS-PAGE should be written with the full name.
12. In line 217, “Figure 1a-b” should be written as “Fig. 1a-b”.
13. In line 254, “cytopathic effects” need to be written in abbreviated form, such as CPE.
14. In line 267, “Q145, Q148, Q159, Q162, and Q196” should be written as “Q145, Q148, Q159, Q162, and Q196”.
15. In lines 275 and 305, “Western blot” should be written as “western blot”.
16. In line 357, “* indicates” need to check for any errors.
17. The content of lines 455 to 464 needs to be deleted.
18. The content of lines 640 to 642 needs to be deleted
19. The images in the manuscript are not aesthetically pleasing, suggestions for improvement.
Reviewer 2 Report
Comments and Suggestions for Authors
The authors demonstrated that the main protease of MERS-CoV induces cleavage of MAVS to antagonize the innate immune response. The experiments are generally well designed.
Figure 1 shows that MAVS is cleaved in MERS-CoV-infected cells. The results are convincing.
Figure 1 comments:
Fig 1 a: please clarify if M means M proteinase?
Fig 1c: please explain in the figure legend nsp5 indicates Mpro
Please keep the abbreviation consistent through the figures and the context: Mpro or Mpro.
Figure 2: The authors identified Mpro cleaved the MAVS, however did not identify Mpro cleavage site.
Figure 3: The authors used pan-caspase inhibitor, Z-VAD, proteasome inhibitor, MG132 and autophagy inhibitor wortmannin confirmed that MAVS cleavage is not caused by apoptosis, proteasome mediated degradation, and autophagy.
Minor comments:
The Figure 3C blot image is poor, it is hard to separate the lanes. If it is possible please provide a better quality image of this blot.
Please provide description for Figure 3d LC3B.
Figure 4 comments:
Please provide description for the first bar of Figure 4a, 4b 4c and 4d.
Figure 5 demonstrates that the cleavage of MAVS is only induced by MERS-CoV, not by SARS-CoV or SARS-CoV2.
References comments:
Please delete the paragraphs between lines 455 – 464.
Please keep the format consistent. For example: Ref 1, the journal name is an abbreviation; Ref 6, the journal name is in full.
Comments on the Quality of English Language
English language and style are fine/minor spell check required
Reviewer 3 Report
Comments and Suggestions for Authors
Dr. van Huizen with colleagues presents an interesting draft focusing on an innate escape of viruses via MAVS-dependent pathway.
I went over the draft with the great interest, since this area lies very close to my scientific interests. I found the draft well-written, with very good structure and high-quality of the data presentation. I also appreciate the quality of western data since sometimes it is pretty hard to get low background when using high concentration of abs, like herein.
I found some very major problems within the paper, they must be addressed/fixed during the peer-review process. Please follow them:
1) The first thing is relatively low novelty of the proposed draft. Both viruses were elegantly described as a players in MAVS-dependent immune escape. There are many papers discussing in details how it comes. Please follow very high impact papers e.g.:
-DOI: 10.1073/pnas.2123208119 or doi.org/10.1038/s41556-021-00712-y
The fact of existence of very solid and well balanced literature data already published within the last 3-4 years drastically reduces the novelty of the presented findings.
2) The second alarming thing is the fact of cleaved MAVS sizes presented in western data. In general, based on my expertise, MAVS is present (when talking about human) as 75kDa and ~55kDa protein. In hepatic cells (which were used herein) its activation leads to dimerization of its structure, not cleavage. Please refer to genuine paper from Lamarre Lab (doi: 10.1128/JVI.01659-08). The very high concentration of used abs might led to detection of partially degradated, not cleaved, forms of MAVS. Moreover, the used abs don't detect any kind of cleaved MAVS. What if observed weak bands of "cleaved" protein are just non-specific products of degradation? Recent papers show that we should focus on dimerization,not cleaved of MAVS (or even better on co-I of MAVS and RIG-I)
3) The used cell line choice is an enigma to me. Why human lung–derived A549 and/or primary nasal cells have not been used?
4) The Authors omitted the key proteins of the presented/analyzed pathway, namely: TBK1/pTBK1, TRIM25 with K63-specific ubiquitination as well as the broad role of IKK family.
5) I don't agree that cells from a single culture we could call "system". Also the meaning of "an overexpression system" is misleading since we are talking only about thousands of same cells from the same line.
6) I also suggest to tone down the conclusions. We don't know which particular mechanism drives observed changes.
7) What is a rationale for presentation on the Fig3C ubiquitination?
8) Please remove from the Reference section Instructions for Authors.
9) I fully agree that the presence of cleavage must be further investigated. This is very risky thesis with very little proof from the point of the principles of molecular biology.
Nevertheless, I appreciate very nicely written paper, focusing on very important medical matter as well as trying to investigate still enigmatic innate immune pathway.
Reviewer 4 Report
Comments and Suggestions for Authors
The manuscript by van Huizen and colleagues has proposed a new way for MERS-CoV to deregulate the host immune system through Mpro-mediated cleavage of MAVS, an important mitochondrial immunoregulatory protein. The authors employed mutated forms of the viral protein to prove their point, demonstrated direct physical contact between the putative partners and addressed cell-mediated mechanisms that could have accounted for their observed effect. In my opinion, they did a fine job to provide as much evidence as possible towards their original hypothesis, despite the fact that they were not able to pinpoint the specific cleavage site on MAVS amino acid sequence. I would only suggest one minor point added in the manuscript for clarity purposes.
Material and Methods:
Line 112: Please provide a reference or a description for the plaque assay method.
Author Response
We kindly thank the reviewer for his/her positive words. We have now added a reference for the plaque assay method in line 119 (former line 112) and numbering of all subsequent references has been updated.
Reviewer 5 Report
Comments and Suggestions for Authors
Coronaviruses (CoVs) counteract host innate immune response to modulate their pathogenesis. Knowledge of the underlying molecular mechanisms is crucial to develop novel antiviral strategies. This manuscript describes the role of MERS-CoV main protease (Mpro) in the modulation of MAVS activity. The manuscript is well-written, the rationale is clear, and the experiments are justified. However, there are a few issues than could be considered to improve the manuscript.
Specific comments:
1. As indicated by the authors, since Mpro is an essential viral protein, performing mechanistic analyses in the context of viral infection is very challenging. This justifies that most of the experiments in the manuscript were over-expression studies. However, there are some additional experiments that would be considered to strength authors’ conclusions:
(i) From Fig. 4 it seems that the effect of MAVS cleavage in IFN response is higher than in NF-kB mediated response. On the other hand, MERS-CoV infection of Huh-7 cells does not completely mimic the innate immune response observed in vivo (i.e., in animal models). Then, is MAVS cleavage observed in other cell types, such as MRC-5 (or even in vivo), where MERS-CoV infection induces IFN response?
(ii) Fig. 2C. Do the anti-nsp5 and/or anti-MAVS antibodies work in immunoprecipitation? That being the case it would be good to include some co-immunoprecipitation experiments of the endogenous proteins in infected cells.
2. Lines 128-129. Why were Mpro tags located differently for MERS-CoV (C-terminal) and for SARS-CoV or SARS-CoV-2 (N-terminal)? Could this affect Mpro activity on MAVS cleavage? How was the protease activity of SARS-CoV and SARS-CoV-2 proteins confirmed?
3. Fig. 1C, 1D and lines 223-229: (i) Please note that the pattern of MAVS in the panels from infected cells are quite different between Figs. 1C and 1D, why? (ii) mitochondria morphology is not appreciated in the images (Fig. 1D), please clarify; (iii) including co-localization coefficients would help to support the conclusions
4. Fig. 4. Including as an additional control other innate immune response inducer (such as RIG-I) would help to support that MERS-CoV Mpro effect on innate immune response is mediated by MAVS cleavage and not cleavage of other proteins in the pathway.
Minor comments:
1. Line 274 and throughout the whole manuscript. Please consider that “CA” seems a bit misleading to define the catalytically inactive protein.
2. Lines 371-373. A point for discussion: although Mpro sequence is only 50% identical, maybe comparison of MERS-CoV and either SARS-CoV or SARS-CoV-2 Mpro structures would help to explain the different behavior and/or the different host substrate (MAVS vs. RIG-I).
Reviewer 6 Report
Comments and Suggestions for Authors
A potential immune escape pathway is addressed in this manuscript. The work is well designed, carried out and presented. It should be published as it is.
Author Response
We kindly thank the reviewer for his/her positive appraisal of our manuscript.
Round 2
Reviewer 3 Report
Comments and Suggestions for Authors
The Authors in a very elegant manner addressed my concerns, modified text accordingly, and discussed my doubts in a very fair fashion. The Authors are aware of the limitations as well as strong points of the study. The overall quality of the paper is very satisfactory, and the way of data presentation is decent.
The last thing which is remaining is to put 2-3 sentence long limitation section at the end of the discussion. Since it is done - the paper is good to go to the production stage.
Author Response
We thank the reviewer for the positive feedback on the way we addressed his/her concerns and revised the manuscript and for the acknowledgement that the overall quality of the paper is very satisfactory.
In the discussion we have extensively discussed several limitations of the study, including the fact that we could not identify the cleavage site (lines 406-418), the need for uncleavable mutants of MAVS so that the effect on downstream signaling can be studied (lines 457-463), and the fact that it is not possible to inhibit Mpro during virus infection, which would be required to further study the role of Mpro in cleavage of MAVS during of infection (lines 448-457). For extra clarity, we have highlighted these sections in yellow. Furthermore, we have added to following sentence in lines 463-465: “Future studies should investigate the effect of Mpro-mediated cleavage of MAVS on the activation and oligomerization of MAVS.”, since this was also an important concern of the reviewer in the first round.